# WWOX Modulates ROS-Dependent Senescence in Bladder Cancer

**DOI:** 10.3390/molecules27217388

**Published:** 2022-10-31

**Authors:** Ching-Wen Liu, Po-Hen Chen, Tsan-Jung Yu, Kai-Jen Lin, Li-Ching Chang

**Affiliations:** 1Department of Senior Citizen Health Service and Management, Yuh-Ing Junior College of Health Care and Management, Kaohsiung 80776, Taiwan; 2Department of Medical Research, E-Da Hospital, Kaohsiung 82445, Taiwan; 3Department of Urology, E-Da Hospital, I-Shou University, Kaohsiung 82445, Taiwan; 4Department of Pathology, E-Da Hospital, I-Shou University, Kaohsiung 82445, Taiwan; 5School of Medicine for International Students, I-Shou University, Kaohsiung 82445, Taiwan; 6Department of Pharmacy, E-Da Hospital, I-Shou University, Kaohsiung 82445, Taiwan

**Keywords:** WWOX, senescence, reactive oxygen species, bladder cancer

## Abstract

The tumor-suppressor gene, WW domain-containing oxidoreductase (WWOX), has been found to be lost in various types of cancers. ROS result as a tightly regulated signaling process for the induction of cell senescence. The aim of this study was to investigate the role of WWOX in the regulation of ROS and cell senescence, which is intriguing in terms of the possible mechanism of WWOX contributing to bladder cancer. In this study, we used the AY-27 rat bladder tumor cell line and F344 orthotopic bladder tumor models to reveal the pro-senescence effects of WWOX and the corresponding underlying mechanism in bladder cancer. WWOX-overexpressing lentivirus (LV-WWOX) remarkably stimulated cellular senescence, including increased senescence-associated secretory phenotype (SASP) formation, enlarged cellular morphology, and induced SA-β-Gal-positive staining. A further mechanism study revealed that the pro-senescence effect of LV-WWOX was dependent on increased intercellular reactive oxygen species (ROS) generation, which subsequently triggered p21/p27. Moreover, LV-WWOX significantly inhibited the tumor size by 30.49% in the F344/AY-27 rat orthotopic model (*p* < 0.05) by activating cellular senescence. The expression of p21 was significantly enhanced in the orthotopic bladder tumors under WWOX treatment. The orthotopic bladder tumors in the groups of rats verified the effect in vivo. Our study suggests that WWOX, an ROS-dependent senescence-induced gene, could be further studied for its therapeutic implications in bladder cancer.

## 1. Introduction

Bladder cancer is the ninth most common urological malignancy worldwide. Approximately 70–80% of bladder cancer (BCa) patients present with non-muscle-invasive (superficial) bladder cancer (NMIBC) that is confined to the mucosa and is characterized by a generally good prognosis, with a tendency to remain localized [1]. However, the rate of intravesical recurrence and the progression rates of NMIBC after transurethral resection of the bladder tumor (TURBT) is still as high as 50%. Management of locally recurrent NMIBC is a significant challenge for a number of reasons [2]. At present, transurethral resection followed by immunotherapy (bacillus Calmette–Guerin) and instillations of chemotherapy is the standard management for non-metastatic bladder cancer, while immunotherapy strategies hold promise to improve the outcome of patients afflicted with this malignancy [3].

The WW domain-containing oxidoreductase (WWOX) gene has been reported to be a tumor suppressor. A significant reduction or loss of expression of WWOX has been observed mainly in breast cancer [4], but also in liver [5] and bladder cancers [6]. In point of fact, the contribution of WWOX to pathways involving oxidative stress and aerobic metabolism are well documented, which provides further scientific evidence for the non-classical tumor suppressor characteristics of WWOX. Its ability to facilitate the circumvention of mitochondrial damage-induced glycolysis has been proposed as a possible mechanism for its tumor-suppressor activity. The function of WWOX in promoting cell death in response to TNFα signaling is mediated by ROS [7]. WWOX contributes to oxidative stress, providing an explanation for the non-classical tumor-suppressor behavior of WWOX [7].

Cellular senescence is a state of permanent cell cycle arrest that can result in the accumulation of ROS exceeding the scavenging capacity of the antioxidant system, which may disturb the redox balance, resulting in oxidative stress [8]. A recent study revealed that cellular senescence can trigger DNA damage, consequently repressing the proliferation and growth of bladder cancer cells, suggesting senescence as a potent tumor suppressor mechanism [9]. Senescent cells actively communicate with neighboring cells and the extracellular matrix through the senescence-associated secretory phenotype (SASP). Subsequently, neighboring cells display high oxidative damage and activation of p53, p16^INK4a^, and p21^cip1^ [10].

However, whether WWOX can trigger ROS generation in bladder cancer cells is unknown, and whether WWOX-inducing senescence depends on ROS generation also remains unclear. The present study aimed to determine the mechanisms of the pro-senescence of WWOX in AY-27 cells in vitro and in an F344/AY-27 rat orthotopic model in vivo.

## 2. Results

### 2.1. Establishment of Cell Lines Overexpressing WWOX

The increased expression of WWOX in lentivirus-infected cells was measured at 48 h post-transfection by Western blot analysis (Figure 1a). As shown in Figure 1b, the WWOX protein expression levels in the AY-27 cells infected with WWOX-overexpressing lentivirus (LV-WWOX) was 2.01-fold (*p* < 0.01) higher than those in the AY-27 cells infected with the lentivirus-containing control vector (LV). The WWOX protein expression levels in the AY-27 cells infected with WWOX-overexpressing lentivirus (LV-WWOX) was 2.49-fold (*p* < 0.05) higher than those in the AY-27 cells (control).

### 2.2. WWOX Suppressed the Proliferation and Migration of Bladder Cancer Cells

To explore the mechanisms of WWOX in cell proliferation, WWOX was overexpressed by stable transfection of the WWOX plasmid construct into the AY-27 cell line expressing the WWOX protein. In the AY-27 cells infected with the WWOX, the cell growth (Figure 2a) showed that there was a significant decrease in number when compared to the cells infected with a control or the LV group cells. 

Cell proliferation was measured using MTS assay. A cell proliferation assay was performed to examine the effect of WWOX on bladder cancer cell growth. LV-WWOX induced a significant suppression in cell proliferation in the AY-27 bladder cancer cell line (* *p* < 0.05; Figure 2b).

Furthermore, a wound-healing assay was performed to assess the effect of WWOX on bladder cancer cell migration (Figure 2c). LV-WWOX significantly decreased the extent of wound healing (percentage of wound closure) in the AY-27 cell line (* *p* < 0.05). These results indicated that LV-WWOX inhibited the cell growth and metastasis of the bladder cancer cells. The wound-healing progress in the control and LV group cultures was rapid, such that the scratch had disappeared in 36 h, while in the treated samples, WWOX inhibited the healing process and the closure of the denuded area progressed more slowly (Figure 2d).

### 2.3. Overexpression of WWOX Increased the TNF and ROS Activity

To determine if WWOX expression may affect the modulation of TNF-α, we performed Western blotting and immunofluorescent staining to determine the changes in TNF-α protein expression (Figure 3). The expression of TNF-α in those AY-27 cells subjected to LV-WWOX treatment was assessed using Western blot analysis (Figure 3a). As shown in Figure 3b, the TNF-α protein expression levels in those AY-27 cells infected with LV-WWOX were 2.17-fold (*p* < 0.001) higher than those in the LV. Those AY-27 cells infected with LV-WWOX were 5.07-fold (*p* < 0.01) higher than the control. The WWOX protein expression levels in those AY-27 cells infected with LV-WWOX were 3.25-fold (*p* < 0.001) higher than those in the LV. Those AY-27 cells infected with LV-WWOX were 3.03-fold (*p* < 0.01) higher than the control.

WWOX alone was transiently expressed in the AY-27 cells through LV-WWOX. Localization of the WWOX and TNF-α was then determined by immunofluorescent staining, as shown in Figure 3c, WWOX or TNF-α localized in the cytoplasm. These results indicate that overexpression of WWOX enhances TNF-α protein expression as shown in Figure 3d, the TNF-α protein expression levels in those AY-27 cells infected with LV-WWOX were 3.29-fold (*p* < 0.01) higher than those in the LV. Those AY-27 cells infected with LV-WWOX were 4.25-fold (*p* < 0.01) higher than the control. The WWOX protein expression levels in those AY-27 cells infected with LV-WWOX were 7.96-fold (*p* < 0.01) higher than those in the LV. Those AY-27 cells infected with LV-WWOX were 9.69-fold (*p* < 0.01) higher than the control.

TNF-α may stimulate ROS production via several sources. Several reports have also demonstrated that TNF-α triggers several signal transduction pathways to activate NOX activity and enhance intracellular ROS generation, leading to the expression of inflammatory genes. Therefore, we investigated whether WWOX-induced TNF-α expression is due to activation of ROS generation. Here, we found that WWOX markedly induced superoxide and hydrogen peroxide production, determined by using DHE under a fluorescence microscope (Figure 3e). To test whether ROS are regulated by WWOX, we measured the ROS levels in response to its overexpression using lentivirus containing WWOX (LV-WWOX). As shown in Figure 3f, the ROS expression levels in those AY-27 cells infected with LV-WWOX were 6.93-fold (*p* < 0.001) higher than the control. The ROS expression levels in those AY-27 cells infected with LV-WWOX were 2.87-fold (*p* < 0.01) higher than those in the LV.

### 2.4. WWOX Induced Premature Senescence in the Bladder Cancer Cells

Cellular senescence is regarded as a physiological barrier against carcinogenesis and tumor progression. In this study investigating the question of suppressing tumor outgrowth, we examined whether these cells underwent senescence. Because senescence-associated β-galactosidase (SA-β-gal) activity is a well-established biomarker of senescence, we investigated if WWOX treatment induces premature senescence in AY-27 cells by SA-β-gal staining (Figure 4a). To test whether senescence is regulated by WWOX, we measured the SA-β-gal levels in response to its overexpression using lentivirus containing WWOX (LV-WWOX). As shown in Figure 4b, the senescence expression levels in the AY-27 cells infected with LV-WWOX were 6.90-fold (*p* < 0.001) higher than the control. The senescence levels in the AY-27 cells infected with LV-WWOX were 3.31-fold (*p* < 0.01) higher than those in the LV. To further demonstrate that WWOX-induced senescence is caused by ROS, we treated cells with Exendin-4 that reduces ROS production. As shown in Figure 4c, Exendin-4 suppressed the WWOX-induced senescence in LV-WWOX group of AY-27 cells in comparison with Figure 4a.

WWOX-regulated cellular senescence occurs through the alteration of gene expression; we analyzed the gene expression of senescence markers. WWOX is activated by multiple stresses, including oxidative stress [11], and the regulation of WWOX has been implicated in cellular senescence by the induction of p21 and p27. A previous study revealed that ROS contributes to cell senescence by the activation of the p21 pathway [12]. Thus, to further determine the mechanism of WWOX-induced senescence and ROS generation, p21 and p27 were evaluated. Western blot analysis was used to identify the increased expression of p21 and p27 in lentivirus-infected cells at 48 h post-infection (Figure 4d). The p21 protein expression levels in those AY-27 cells infected with LV-WWOX were 2.14-fold (*p* < 0.05) higher than those in the LV. Those AY-27 cells infected with LV-WWOX were 2.57-fold (*p* < 0.01) higher than the control. The p27 protein expression levels in those AY-27 cells infected with LV-WWOX were 1.64-fold (*p* < 0.01) higher than those in the LV. Those AY-27 cells infected with LV-WWOX were 1.41-fold (*p* < 0.05) higher than the control.

### 2.5. WWOX Suppressed Bladder Cancer Growth in the F344/AY-27 Rat Model

In order to further determine the antitumor effect of WWOX-induced senescence, an F344/AY-27 rat orthotopic model was established. The AY-27 rat bladder tumor cells were introduced into the bladders of F-344 rats. In control and LV groups, the rats developed gross hematuria and a palpable suprapubic mass. Notably, WWOX suppressed tumor growth (Figure 5a) and tumor size (Figure 5b). Meanwhile, compared to LV-WWOX, the control and LV also significantly decreased the tumor size by 30.49 and 28.60%, respectively (*p* < 0.05).

The control group exhibited bladder tumor invasion from the urothelium to the muscle layer of stage T2. The tumor cells exhibited an increased variation in nuclear size and mitotic figures. The LV group also showed similar histologic features of bladder tumors, including marked nuclear pleomorphism. In the LV-WWOX group, low-to-moderate dysplasia and a concentration of inflammatory cells were evident, and relevant progress to advanced stage (T1 stage). The LV-WWOX (magnified view) indicated increased polymorphonuclear leukocyte infiltration in the tumor, observed as dark purple-stained granules in the connective tissue beneath the bladder lining (T1 stage). Several tumor nests were apparent with cell debris confined inside. The images illustrate the anticancer activity of LV-WWOX in the bladder (Figure 5c).

However, as to whether the antitumor effect of WWOX can be ascribed to senescence induction, we analyzed p21 expression via immunohistochemistry (IHC) in rat orthotopic bladder cancer tissue samples. p21 was highly expressed in the LV-WWOX group compared to the control and LV group (Figure 5d). These results suggest that WWOX could suppress bladder tumorigenesis by stimulating cellular senescence.

## 3. Discussion

WWOX is a gene spanning the FRA16D common chromosomal fragile site. WWOX functions as a tumor-suppressor gene in a number of cancers, including both breast and bladder cancer [13,14,15,16,17], but the underlying functional mechanisms of WWOX have not been fully explained. In the present study, we demonstrated that expression of WWOX in bladder cancer AY-27 cells suppresses proliferation. The modulation of WWOX in bladder cancer cells has a dramatic effect on cell morphology and proliferation (Figure 1). In addition, we found that WWOX strongly activates the inflammatory response and ROS generation in the AY-27 bladder cancer cell line. The compelled expression of WWOX produced significant morphologic changes, induced ROS, and profoundly inhibited cell proliferation. ROS can lead to DNA damage and can cause DNA mutations. These DNA mutations lead to the activation of tumor-suppressor genes, inhibiting tumorigenesis by decreasing tumor cell proliferation, migration, and resistance to apoptosis [18]. Because fragile site gene expression is absent in most cancers [19], most cancers have lost this important ROS modulator, allowing the inappropriate survival and growth of cancers.

WWOX contributes to pathways involving oxidative stress, providing an explanation for the non-classical tumor suppressor. Its ability to facilitate the circumvention of mitochondrial damage-induced glycolysis (Warburg effect) has been proposed as a possible mechanism for its tumor-suppressor activity [11]. Previous studies have indicated that WWOX induction elicits the production of a variety of cytokines, including TNF-α and IL-1β [20]. Immunohistochemical analysis showed that the expression of both green fluorescent protein-tagged WWOX and red fluorescent protein-tagged TNF-α in the AY-27 cells also resulted in cytosolic colocalization (>80% in TNF/WWOX-expressing cells) (Figure 3d).

Interestingly, in addition to growth suppression, WWOX overexpression also induced morphological changes that are characteristic of cellular senescence. To the best of our knowledge, the present study is the first that correlates WWOX with cellular senescence in urothelial carcinoma. In our study, the antitumor effect of WWOX was determined and the mechanism involved in senescence was subsequently evaluated. Moreover, WWOX-induced senescence is closely dependent on intercellular ROS accumulation, supporting the induction of cellular senescence in bladder cancer cells after WWOX overexpression. Because the primary role of cellular senescence is considered to be cancer prevention, the induction of senescence may be useful for the treatment of cancer [21].

This study examined the optimal gene delivery system for the intravesical therapy of bladder cancer to improve the efficacy and reduce the adverse effects in a rat model. Intravesical therapy is an attractive method because it minimizes systemic exposure to drugs by selectively delivering drugs to the bladder tissues containing tumors [22]. An orthotopic bladder tumor model is indispensable in studies of therapeutic effects. AY-27 rat bladder cancer cells transplanted orthotopically into Fischer F344 female rats exhibit features of urothelial carcinoma [23]. Patchy carcinoma in situ can be detected histologically at 7–10 days after inoculation, and the condition progresses to papillary tumor or invasive disease thereafter (Figure 5a). Moreover, no pathologic changes were observed in the organs (heart, liver, and kidney) (data not shown), suggesting no observed systemic toxicities were induced by lentivirus. Intravesical LV-WWOX instillation exhibited its anticancer effect on urothelium carcinoma that had been orthotopically implanted in a rat model.

Taken together, by determining the pro-senescence mechanism of WWOX in bladder cancer cells, the present study advances our understanding of ROS-induced senescence and the antitumor effect of WWOX in the following aspects: (i) ROS-dependent senescence, not relying on oncogene activation, is partially restricted from oncogene-induced senescence, which implies that there may be some underlying alternative mechanisms that need extensive investigation; (ii) p21 activation could promote ROS-induced senescence; (iii) WWOX may predominantly stimulate ROS-dependent senescence to eliminate the senescent tumor cells.

## 4. Materials and Methods

### 4.1. Virus Production and Incorporation with LV-WWOX

Lentivirus was prepared for the studies using established techniques. Lentivirus was produced in 293T cells grown in Dulbecco’s modified Eagle’s medium plus 10% fetal bovine serum at 37 °C and 5% CO_2_. The lentiviral packaging vectors (pMDL-GagPol, pRSV-Rev, and pIVS-VSV-G) were co-transfected, along with pLKO_AS2.puro (LV) or pLKO_AS2.puro-WWOX (LV-WWOX), into 293T cells using TranslT-LT1 (Mirus, Madison, WI, USA). After 48 h of transfection, the supernatant was collected and filtered (0.45 micron). The infectious titer (infectious unit (IU)) of LV-WWOX was determined by counting the number of expressed WWOX cells two days after incubation of serially diluted viruses with 293T cells.

### 4.2. Lentivirus Release

Release studies were performed by incubation of lentivirus in serum containing media at 37 °C in 293T cells, with samples (100 μL) collected via replacement at the indicated time points and stored at −80 °C until the sample concentration was determined. Viral particle concentrations were determined using HIV-1 p24 Antigen ELISA (ZeptoMetrix, Franklin, MA, USA) [24].

### 4.3. Cell Culture and Cytotoxic Assay

The rat bladder cancer cell line AY-27 (a kind gift from Professor R. Moore, University of Alberta, Edmonton, AB, Canada) was cultured in RPMI-1640 medium supplemented with 10% fetal bovine serum (FBS), 2% L-glutamine, and 0.2% penicillin/streptomycin at 37 °C and 5% CO_2_. Cell viability was determined using the CellTiter 96 Aqueous nonradioactive cell proliferation assay according to the manufacturer’s instructions (Promega, Madison, WI, USA). AY-27 cells (1 × 10^4^) were seeded in 96-well plates and then treated with epirubicin-loaded hydrogels for 18 h. The cell growth inhibition potency is expressed as IC_50_ values, defined as the concentration of the drug necessary to inhibit the growth of cells by 50% in 18 h. The percentage of surviving cells was defined as (mean absorbance of treated wells/mean absorbance of untreated wells) × 100%. The data are the means ± SDs of four experiments. 

### 4.4. Cell Migration Test

The cells were seeded in six-well plates and cultured. A line was created by scraping the cells with a sterile pipette in the middle of the well. The wound area was marked and cell migration into the scratched area was photographed daily using an inverted microscope equipped with a camera (Nikon) for two days. The wound area was calculated by manually tracing the cell-free area in the captured images using the ImageJ public domain software (NIH, Bethesda, MD, USA). Under normal conditions, the wound area decreases over time. The migration rate can be expressed as the change in the wound area over time [25,26].

### 4.5. Immunocytochemistry and Western Blotting

Immunocytochemistry experiments were performed following a protocol as described previously. The cells were perforated with 0.2% triton-100 in PBS (PBS-T). After blocking with antibody diluent reagent (DAKO, Glostrup, Denmark), the cells were incubated with primary antibody WWOX (Imgenex, San Diego, CA, USA) and TNF-α (BioLegend, San Diego, CA) in antibody diluent reagent (DAKO, Glostrup, Denmark) for 1 h at room temperature (RT). A secondary antibody was applied to the antibody diluent reagent (DAKO, Glostrup, Denmark) for 1 h at RT. The stained tissues were mounted with shield mounting medium (DAKO, Glostrup, Denmark) and photographed using the Nikon Eclipse Ti inverted microscope. The AY-27 cells were washed twice in cold PBS and lysed in TBS containing 1 mM EDTA, 1 mM DTT, 0.2% Triton, 0.1% SDS, and a complete protease inhibitor mixture (Roche, Indianapolis, IN, USA). The proteins were subjected to SDS-PAGE and then transferred to polyvinylidene difluoride membranes (Boehringer Mannheim, Germany). The membranes were probed with antibodies specific for WWOX (Imgenex, San Diego, CA, USA), TNF-α (BioLegend, San Diego, CA, USA), or β-actin (Merck Millipore, Darmstadt, Germany), and incubated with HRP-conjugated anti-rabbit Ig or anti-mouse Ig Ab. Signals were revealed with an ECL kit (Merck Millipore, Darmstadt, Germany) and visualized by autoradiography.

### 4.6. Measurement of Intracellular ROS

To evaluate the ROS production by AY-27-LV-WWOX, dihydroethidium (DHE) staining was employed according to the manufacturer’s instructions as previously reported. Briefly, the cells were pretreated with interventions and then loaded with DHE (100 μM; Sigma, St. Louis, MO, USA) for 30 min in the dark. The cells were observed via a fluorescence microscope at magnification ×400 (Eclipse 80i, Nikon, Tokyo, Japan), and the positive areas were analyzed using ImageJ 1.42 software.

### 4.7. Senescence-Associated β-Galactosidase Assay (SA-β-Gal)

The cells were seeded 24 h prior to staining at 1 × 10^5^ cells/well in 24-well plates. Briefly, the cells were washed with cold PBS, and fixed for 5 min with 4% glutaraldehyde diluted in cold PBS. After fixation, the cells were washed in PBS and incubated for 8 h at 37 °C in a staining solution containing 1 mg/mL of 5-bromo-4-chloro-3-indolyl-β-D-galactoside (X-Gal) (Roche, Indianapolis, IN, USA). Following the incubation period at 37 °C, the cells were washed 3 × 5 min with cold PBS and stored in PBS at 4 °C until images were collected. Exendin-4 has been reported to reduce ROS production [27]. Exendin-4 (100 nM, (Sigma-Aldrich, St. Louis, MO, USA)-treated AY-27 cells for 24 h were subjected to SA-β-Gal staining.

### 4.8. Ethics Statement and In Vivo Urothelium Permeability and Histologic Analysis

All in-life rat studies were performed under the Association for Assessment and Accreditation of Laboratory Animal Care-approved conditions and approved by the I-Shou University Institutional Animal Care and Use Committee (Approval numbers IACUC-ISU-101027). In situ urothelium cancer was induced using intravesical instillation of the AY-27 line of rat bladder tumor cells.

F344 rats were anesthetized, and the bladder was catheterized via a urethral catheter. To facilitate tumor seeding, the bladder mucosa was conditioned with 0.4 mL of 0.1 N hydrochloric acid (HCl) for 15 s and neutralized with 0.4 mL of 0.1 N potassium hydroxide (KOH) for 15 s. The bladder was then drained and flushed with sterile PBS. Immediately after bladder conditioning, the AY-27 cells (1 × 10^7^) were instilled and left indwelling for at least 1 h. The rats were turned 90° every 15 min to facilitate whole bladder exposure to the tumor cell suspension. After 1 h, the catheter was removed and the rats were allowed to void the suspension spontaneously.

After tumor implantation (on the first day), F344 rat bladders were intravesically instilled with LV or LV-WWOX on the 8th, 10th, and 12th days. All of the rats were sacrificed on the 14th day. On the 14th day, the bladders were excised, fixed in 4% formalin overnight, dehydrated, and then embedded in paraffin. The embedded tissues were then cut into 5 μm slices and stained with hematoxylin and eosin (H&E) for histological analysis.

### 4.9. Statistical Analysis

The data are presented as the mean ± standard deviation (SD). The data were analyzed using the Student’s *t*-test and one-way analysis of variance (ANOVA). Statistical significance was determined at the level of *p* < 0.05.

## 5. Conclusions

In conclusion, WWOX caused cytotoxicity and the induction of senescence in the AY-27 bladder cancer cells. TNF-α and p21 may both play an important role in the prevention of bladder cancer. In the orthotopic bladder tumor model, WWOX was capable of inhibiting tumor growth. To the best of our knowledge, this is the first report showing the antitumor effects of WWOX on bladder cancer. Our study demonstrated the critical role of WWOX in bladder cancer development and suggests that WWOX can be further studied for its therapeutic implications in bladder cancer. WWOX may become one of several promising strategies directed at tumor suppressor genes or related pathways for cancer therapy in the future.

## Figures and Tables

**Figure 1 molecules-27-07388-f001:**
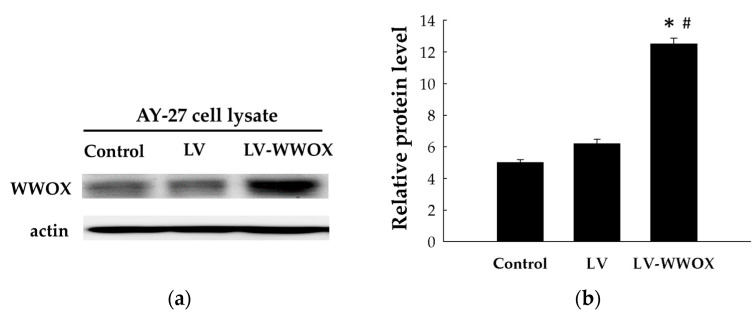
Establishment of WWOX-overexpressing cell lines. (**a**) AY-27 cells were infected with lentivirus overexpressing WWOX or containing the control vector. The total proteins were extracted after 48 h of infection and the concentrations were measured. Subsequently, Western blot analysis was performed to detect the expression of WWOX in the AY-27 cells. (**b**) WWOX protein expression was normalized to actin expression. The results are expressed as the mean ± SD. * *p* < 0.05 compared to the control groups; # *p* < 0.01 compared to the LV groups.

**Figure 2 molecules-27-07388-f002:**
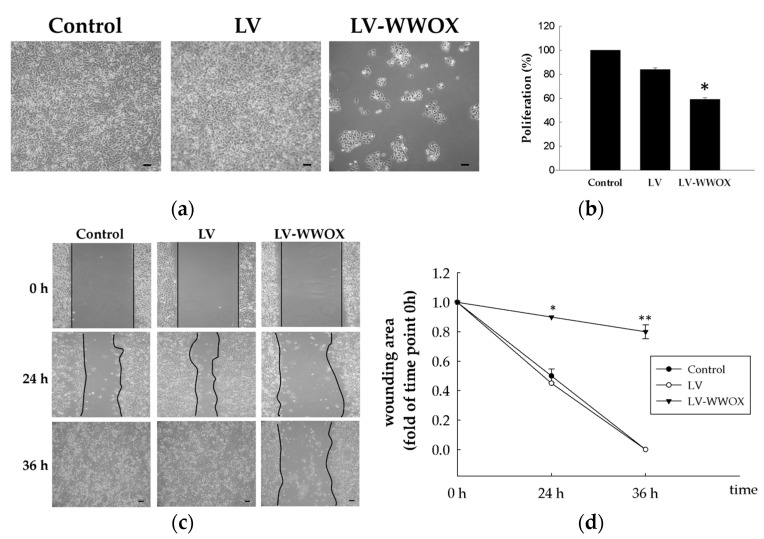
WWOX overexpression in the bladder cells inhibited cell proliferation in vitro. The (**a**) cell growth and (**b**) cell proliferation of the AY-27 cells were measured by MTS assay. (**c**) Wound-healing assay and (**d**) wound closure expressed as the remaining area uncovered by the cells. The scratch area at time point 0 h was set to 1 (*n* = 3; * *p* < 0.05 and ** *p* < 0.01 vs. the respective control condition). The black bar in the right lower corner is 100 mm.

**Figure 3 molecules-27-07388-f003:**
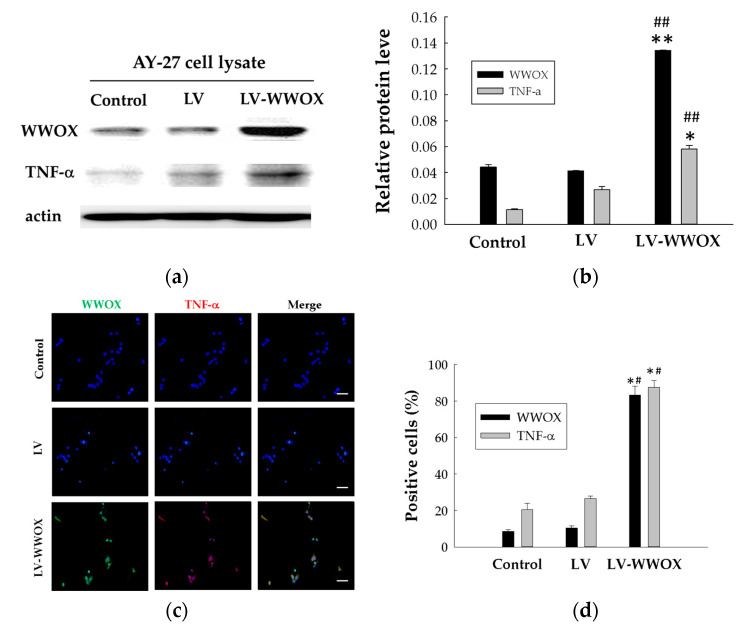
Overexpression of WWOX in AY-27 cells. WWOX-low AY-27 cells were infected with WWOX (LV-WWOX) or the negative control gene lentivirus (LV). (**a**) The expression of WWOX and TNF-α in LV-WWOX was detected by Western blot. Actin served as a loading control. (**b**) Protein expression was normalized to actin expression. The results are expressed as the mean ± SD. (**c**) The localization of WWOX and TNF-α in LV-WWOX was measured by immunofluorescence (×200). (**d**) The levels are expressed in positive cells (%). (**e**) Detection of ROS using DHE fluorescent dye in the AY-27, AY-27-LV, and AY-27-LV-WWOX cells (×200). (**f**) The levels are expressed in ROS-positive cells (%). The white bar in the right lower corner is 100 mm. Each bar represents the mean ± standard deviation (SD) from three independent experiments. The results are expressed as the means ± SDs. * *p* < 0.01 and ** *p* <0.001 compared to the control groups; ^#^ *p* < 0.01 and ^##^ *p* < 0.01 compared to the LV groups. The white bar in the right lower corner is 100 mm.

**Figure 4 molecules-27-07388-f004:**
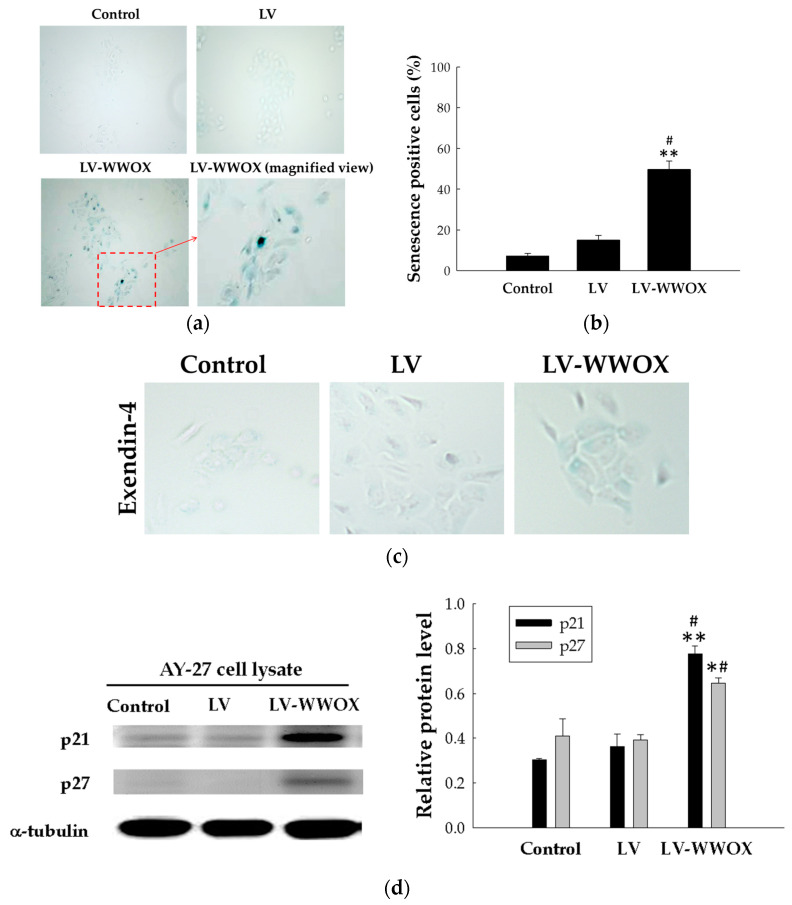
WWOX induced premature senescence in AY-27 cells. (**a**) SA-β-gal staining increased with LV-WWOX treatment in the AY-27 cells. (**b**) The percentage of SA-β-gal-positive senescent cells in the LV-WWOX-treated, LV, and control AY-27 cells is presented as the mean ± SD. ** *p* < 0.01 compared to the control groups; # *p* < 0.05 compared to the LV groups. (**c**) SA-β-gal staining of exendin-4-treated AY-27 (control), AY-27-LV, and AY-27-LV-WWOX cells. (**d**) The expression of p21 and p27 in the LV-WWOX-treated, LV, and control AY-27 cells was detected by Western blot. α-Tubulin served as a loading control. Each bar represents the mean ± standard deviation (SD) from three independent experiments. The results are expressed as the mean ± SD. * *p* < 0.05 and ** *p* < 0.01 compared to the control groups; # *p* < 0.05 compared to the LV groups.

**Figure 5 molecules-27-07388-f005:**
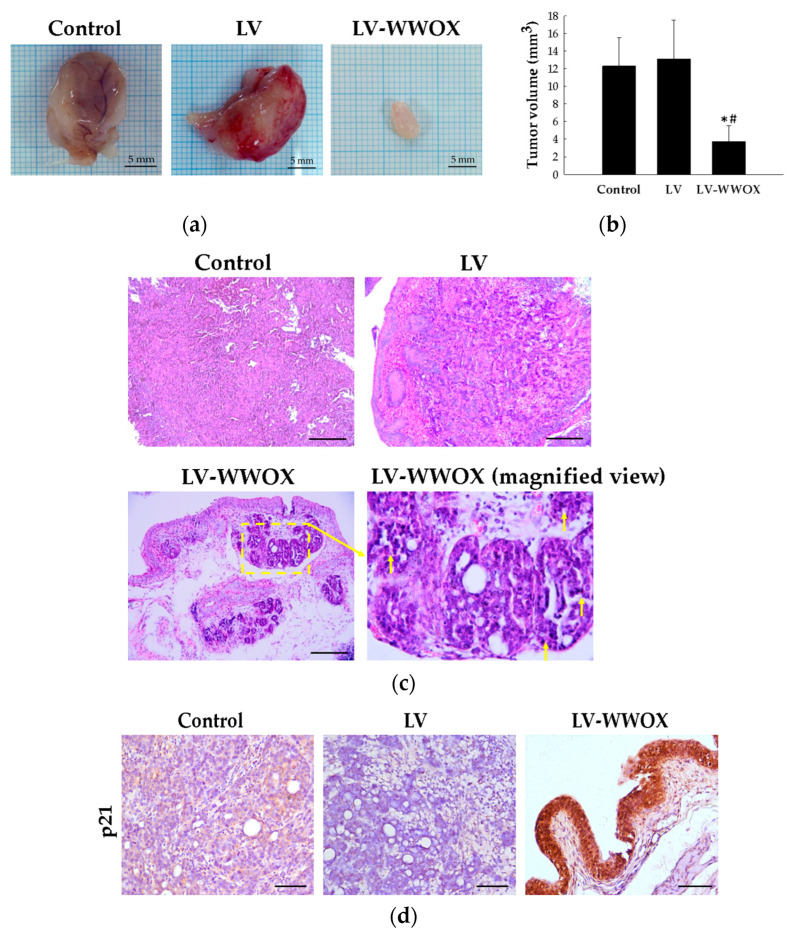
Inhibitory effects of WWOX in the F344/AY-27 rat model. (**a**) Excised tumor photographs. (**b**) Tumor volume (tumor volume = (longer diameter × shorter diameter^2^)/2) was analyzed. The results are expressed as the mean ± SD. * *p* < 0.05 compared to the control groups; # *p* < 0.05 compared to the LV groups. (**c**) Histopathologic findings with H&E staining. (**d**) Images of the tumor p21waf protein expression presented by immunohistochemistry (scale bar = 100 μm). Original magnification was 400×.

## Data Availability

The raw data will be available from corresponding author upon reasonable request.

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
