# Peer review of "WWOX Modulates ROS-Dependent Senescence in Bladder Cancer"

_molecules, 2022, doi:10.3390/molecules27217388_

Round 1

Reviewer 1 Report

Liu et al. demonstrate that exogenous overexpression of WWOX leads to upregulated ROS and subsequent senescence induction, conferring a tumor maintenance benefit in vivo. However, the currently submitted article has very minimal mechanistic experiments and should be revised significantly. 

1. Lines 63-64, the authors suggest that the SAHF is responsible for inter-cellular communication between senescent cells and others; this is wrong. It is the SASP (senescence-associated secretory phenotype). 

2. This reviewer does not see the relevance or importance of figure 1C. Why the authors included this needs to be better explained. 

3. The authors need to better explain the rationale for looking at TNFalpha in the results section. It comes rather abruptly and does not seem to fit with the rest of the "story". 

4. For figure 3, was TNFalpha exogenously expressed with lentivirus as suggested in line 147 of the manuscript? If so, which conditions shown are WWOX, TNFa, and WWOX+TNFa over-expressing in the figure? This is not clear. Also, the rationale for overexpressing TNF alpha is unclear and this reviewer is unconvinced of the value of these experiments. 

5. The authors suggest that TNF-alpha also affects WWOX expression and localization (lines 150-151). However, as indicated in point 4, it is unclear in figure 3 which conditions have TNF-alpha overexpressed. Further, while this reviewer agrees that TNF-alpha and WWOX clearly co-localize to the same cells, it is hard to see whether WWOX is localized differently based on TNFa expression with the currently provided images. 

6. Is there supposed to be a TNFa mechanistic link to the ROS data? If so, there is no data supporting said mechanism, and if not, then again, this reviewer believes the TNFa data is irrelevant to this manuscript. 

7. Figure 3's legend is written incorrectly. 

8. Figure 4C- the cropping is inappropriate. The bands of interest are almost cut off. Please expand the cropped region so that there is some buffer room on all sides of the band. 

9. The authors are encouraged to do scavenger experiments to demonstrate that the accumulation of ROS is indeed driving WWOX induced senescence, rather than other effects of WWOX. 

10. The authors suggest an increased concentration of inflammatory cells were seen in the LV-WWOX animal group (lines 239-240), but where is the supporting data for this statement?

11. Do the authors have tumor volume over time to include rather than ending volume? 

12. The authors state on lines 295-297 "We found that rather than inducing apoptosis, WWOX predominantly stimulated cellular senescence via triggering DNA damage of p53/p21". The authors have shown no data about DNA damage or p53. p21 induction does not always indicate DNA damage, so it cannot be inferred that DNA damage has occurred. 

13. The authors are encouraged to discuss the intravesical delivery in the results section a bit more to help contextualize the data more. 

14. Lines 321-322 "WWOX may predominantly stimulate ROS-dependent senescence, while simultaneously inhibit cell proliferation to eliminate the senescent tumor cells". How does inhibition of cell proliferation eliminate senescent cells? This does not make sense. 

15. The references can surely be expanded. 

16. The authors should reference Gang Li et al. "Ectopic WWOX expression inhibits growth of 5637 bladder cancer cell in vitro and in vivo" 2015. 

17. Was the migration/scratch wound assay performed in serum-free media as is standard? Otherwise the difference in closure is easily explained by the growth rate differences rather than true migration.  

Reviewer 2 Report

The author has presented significant research work related to the elucidation of prognostic biomarkers in bladder cancer . However, there is a strong need to improve the language throughout the manuscript.

1. Abstract needs to be reframed with proper scientific terminologies

2. language need to be improved in the manuscript

3. In figure 1a, the blotting image is not clear

4. Author is suggested to add a paragraph related to the other potential biomarkers that have been explored in bladder cancer. This would help in correlating your research work in terms of potential prognostic biomarkers. You can cite this research article as well as other research articles related to these biomarkers in bladder cancer

Pandey, P., Bajpai, P., Siddiqui, M. H., Sayyed, U., Tiwari, R., Shekh, R., ... & Kapoor, V. K. (2019). Elucidation of the chemopreventive role of stigmasterol against Jab1 in Gall bladder carcinoma. Endocrine, Metabolic & Immune Disorders-Drug Targets (Formerly Current Drug Targets-Immune, Endocrine & Metabolic Disorders)19(6), 826-837.

Pandey, P., Siddiqui, M. H., Behari, A., Kapoor, V. K., Mishra, K., Sayyed, U., ... & Bajpai, P. (2019). Jab1-siRNA induces cell growth inhibition and cell cycle arrest in gall bladder cancer cells via targeting Jab1 signalosome. Anti-Cancer Agents in Medicinal Chemistry (Formerly Current Medicinal Chemistry-Anti-Cancer Agents)19(16).

5. Please elaborate on the conclusion section by providing the future aspect of your study.

I would recommend the acceptance of this research article after these incorporations.

Round 2

Reviewer 1 Report

The revised manuscript is much improved. Please address the following prior to final publication: 

1. The SAHF needs to be changed to SASP in line 66, not just in the abstract. 

2. The abstract mentions DNA damage; as in the text of the revised manuscript, please remove from the abstract as well 

3. I would suggest including the scavenger experiments provided in the author response in the main figure relating to senescence if possible. 

4. 

Reviewer 2 Report

The manuscript can be accepted in its current form.
